# Phenotypic Characterization and Phylogeny of *Godronia myrtilli* (Anamorph: *Topospora myrtilli*)—Causal Agent of *Godronia* Canker on Highbush Blueberry

**DOI:** 10.3390/pathogens12050642

**Published:** 2023-04-26

**Authors:** Ewa Mirzwa-Mróz, Marek Stefan Szyndel, Mateusz Wdowiak, Marcin Wit, Elżbieta Paduch-Cichal, Anna Wilkos, Karolina Felczak-Konarska, Wojciech Wakuliński

**Affiliations:** 1Division of Plant Pathology, Department of Plant Protection, Institute of Horticultural Sciences, Warsaw University of Life Sciences—SGGW, 159 Nowoursynowska, 02-776 Warsaw, Poland; 2Fertico Sp. z o.o., 43 Goliany, 05-620 Bledow, Poland

**Keywords:** *Godronia myrtilli*, *Godronia cassandrae*, *Topospora myrtilli*, morphology, mycelium growth rate, PCR detection and identification

## Abstract

*Godronia* canker caused by *Godronia myrtilli* (Feltgen) J.K. Stone is considered one of the most dangerous diseases of blueberry crops. The purpose of the study was the phenotypic characterization and phylogenetic analysis of this fungus. Infected stems were collected from blueberry crops in the Mazovian, Lublin, and West Pomeranian Voivodships in 2016–2020. Twenty-four *Godronia* isolates were identified and tested. The isolates were identified on the basis of their morphology and molecular characteristics (PCR). The average conidia size was 9.36 ± 0.81 × 2.45 ± 0.37 µm. The conidia were hyaline, ellipsoid or straight, two-celled, rounded, or terminally pointed. The pathogen growth dynamics were tested on six media: PDA, CMA, MEA, SNA, PCA, and Czapek. The fastest daily growth of fungal isolates was observed on SNA and PCA, and the slowest on CMA and MEA. Pathogen rDNA amplification was performed with ITS1F and ITS4A primers. The obtained DNA sequence of the fungus showed 100% nucleotide similarity to the reference sequence deposited in the GenBank. Molecular characterization of *G. myrtilli* isolates was performed for the first time in this study.

## 1. Introduction

The most commonly observed disease symptoms on the stems of highbush blueberry (*Vaccinium corymbosum* L.) are referred to as blueberry canker [1,2,3], stem blight [2], dieback [3,4,5,6], or twig blight [7]. They can be caused by many different fungal pathogens, and thus the above names are often used with the addition of the generic name of the pathogen: *Godronia* canker [8,9,10,11], *Fusicoccum* canker [9,12,13,14,15], *Phomopsis* canker [6], etc.

The symptoms of the abovementioned diseases on the stems and shoots can be described as ellipsoidal necrotic spots of a chestnut-brown color with a brighter center and red-purple margins. They are located mainly in the lower parts of one- or two-year-old stems [3,16,17].

Studies on the biodiversity of fungi inhabiting highbush blueberry stems in southern Poland have shown the presence of species or genera: *Topospora myrtilli* (Feltgen) Boerema, *Phomopsis archeri* B. Sutton, *Botrytis cinerea* Pers., and *Cytospora* spp. as well as *Phoma*, *Myxothyrium*, and *Seimatosporium*. The species *T. myrtilli* has been recognized as the most common and harmful pathogen of *V. corymbosum* L. [18].

The species *T. myrtilli* and *Fusicoccum putrefaciens* Sher. [19] have been stated to be the conidial stage of *Godronia cassandrae* (Peck) Groves f. *vaccinii* [20,21,22,23,24,25]. However, Groves [20] (according to Strømeng and Stensvand [11]) has stated that *T. myrtilli* was used as an anamorph of *G. cassandrae* earlier than *F. putrefaciens*, therefore the correct name should be *T. myrtilli*. Additionally, the *Fussicoum* genus has been established based on *F. aesculi* Corda, and it is not the conidial stage of *G. cassandrae* (according to the figures and descriptions therein).

Nowadays, according to Johnston et al. [26], Rossman et al. [27], and the Index Fungorum [28], the current name of this fungal species is *Godronia myrtilli* (Feltgen) J.K. Stone, and this name will be used in this article.

Neither the type of disease symptoms (even if the etiological signs of the pathogen are present) nor anatomical examinations of infested stems allow proper and complete diagnosis of the pathogen [29,30]. For full mycological identification of the causal agent, analysis using artificial culture methods as well as molecular biology methods is necessary [31,32,33].

The aim of this study was the phenotypic characterization and phylogenetic analysis of the causal agent of *Godronia* canker disease on the stems of highbush blueberry plants located in the West Pomeranian, Mazovian, and Lublin Voivodships. 

Additionally, the growth dynamics of the selected fungal isolate were measured on different media, and the molecular characteristics of the selected isolates were studied.

## 2. Materials and Methods

### 2.1. Plantation Observations

The investigations were carried out in 2016–2017 on a private highbush blueberry plantation in the West Pomeranian Voivodship and on commercial plantations in 2019–2020 in the Mazovian and Lublin Voivodships, Poland. The total area of the inspected crops was over 50 ha. Monitoring the development of disease symptoms and the appearance of etiological signs was carried out every two weeks from mid-December to the end of October of the next year.

Blueberry (cv ‘Duke’) stem fragments with visible etiological signs were collected and investigated under a dissecting microscope (SZ11, Olympus, Tokyo, Japan). Next, stem fragments with visible pycnidia were rinsed with sterile distilled water (5 min) and kept in a humid chamber (2–5 days). Exuded conidia from the pycnidia were transferred onto a PDA medium in Petri dishes (10 cm dia) and incubated under white light at 20 °C. Based on morphology and an average size of 150 conidia (randomly selected from 24 obtained isolates), identification of the pathogen was performed. The conidia were examined under a BX50 light microscope (Olympus). Measurements of the size of the conidia were made using the CellF program (Olympus).

### 2.2. Koch’s Postulates

To fulfill Koch’s postulates, six plants of the “Bluegold” blueberry were inoculated with six randomly selected (based on morphological features on PDA medium) isolates of *Godronia* fungus. Plant shoots (of a 1.5-year-old plant grown in pots) were rinsed with sterile water and damaged with a sterile scalpel. The inoculum was discs (5 mm dia) cut out with a sterile cork borer from 10-day fungi colonies on the PDA medium. The discs were applied to the wounded sites of shoots and wrapped with sterile cotton wool. The wounds in the control combination were wrapped with sterile cotton only. The experiment was done in three replications. After inoculation, all plants were sprayed with water and kept in large foil bags for two weeks. The development of disease symptoms was monitored. 

### 2.3. The Fungus Morphology and the Pathogen Growth Dynamics

The morphology and the growth dynamics of the fungus representative isolate (PIC2) were determined on six media: CMA (corn meal agar), PDA (potato dextrose agar), MEA (malt extract agar), SNA (synthetic nutrient-poor agar), PCA (potato-carrot agar), and Czapek (Czapek solution agar). Discs (5 mm dia) overgrown with pathogen mycelium were cut out from the PDA medium and placed in the center of Petri dishes with solid media (10 Petri dishes per replication). Next, the dishes were incubated under white light at 20 °C (14 h day/10 h night). Colony size measurements were taken 3, 5, 7, 9, 12, 15, 17, 19, and 21 days after inoculation. The areas of fungal colonies were calculated using the formula for the area of the ellipse.

To determine the dynamics of colony growth, regression analysis was performed. The regression equations for each medium were determined according to the formula: y = bx + a(1)

a—estimated interceptb—estimated slopey—area of the mycelium (mm^2^)x—day of the experiment

The b coefficient of the equation was taken as an average daily growth rate of fungal cultures on a given medium (mm^2^/24 h) [34]. 

A one-way ANOVA was used to check the differences between the b coefficients of each medium. To determine homogeneous groups of the b coefficients, the Student–Newman–Keuls (SNK) multiple comparison procedure was used (at a significance level of *p* = 0.05). Statistical analysis was performed using the Statgraphics Plus 4.1 for Windows 4.1. software (Statgraphics Technologies, Inc., The Plains, VA, USA).

### 2.4. PCR and Phylogenetic Relationship between Godronia Isolates

The rDNA extraction was carried out using a Wizard Genomic DNA Purification Kit (Promega) according to the company’s protocol. The polymerase chain reaction (PCR) analyses of 24 fungal isolates were performed using the primer pair (F) ITS1F and (R) ITS4A [35]. The reaction mixture (50 μL) comprised 2 μL of the extracted DNA, 0.5 μL of Taq Polimeraza, and 0.25 μL each primer. Thirty-six cycles were performed in the Applied Biosystems Veriti 96 Wel Thermal Cycler. PCR amplification was performed with an annealing temperature of 57 °C. The electrophoretic separation of the PCR reaction products was conducted in 1.2% agarose gel in the presence of ethidium bromide by using a horizontal electrophoresis apparatus (Easy-Cast^TM^ Horizontal System model B1.A (Owl^TM^ Separation Systems, Thermo Fisher Scientific Inc., Waltham, MA, USA). 

The sequencing of the PCR reaction product of six selected isolates was performed using Sanger’s sequencing method at the Polish Academy of Sciences, Institute of Biochemistry and Biophysics (Warsaw, Poland). The sequencing results were analyzed using the BLAST algorithm on The National Center for Biotechnology Information (NCBI, https://blast.ncbi.nlm.nih.gov/, accessed on 10 September 2022) database and MEGA version 5 [36]. The DNA sequences were aligned using Clustal W version 2.0 [37].

Due to the small number of nucleotide sequences of the *G. cassandrae* species in the GenBank, the sequences of other species of the *Godronia* genus were also used in creating a dendrogram. A small subunit ribosomal RNA gene, partial sequences; an internal transcribed spacer 1 and 5.8S ribosomal RNA gene, complete sequence; and an internal transcribed spacer 2, partial sequence of *G. myrtilli* from Poland (OP902995) and 13 selected *Godronia* species from the NCBI database were analyzed using the maximum likelihood approach for phylogeny reconstruction and analyzed by bootstrapping with 1.000 replicates. A *Monilinia fructicola* (G. Winter) Honey sequence was included as an outgroup. Missing and ambiguous characters were excluded from the analysis [38].

## 3. Results

### 3.1. Plantation Observations

*Godronia* disease symptoms caused by *G. myrtilli* on highbush blueberry plants were observed on one-year-old stems during the whole winter period starting from mid-December 2016. Infected plants were noticed only on an old private plantation in the West Pomeranian Voivodship. Symptoms occurred mainly in the lower part of the shoots, around leaf scars or buds, as ellipsoidal necrotic spots of a chestnut-brown color with a brighter, gray-ashen center and a purple edge. Leaf scars, wounds, and cracks were the most likely sites of new infections. The conidial stage in the form of the pycnidia was found either under the epidermis or, after its breaking, on the surface of the stems from mid-March to June. Later, the shoot usually died out above the site of the infection, but the first signs of shoot wilting were noted from July to late summer. For every five blueberry bushes, two or three shoots died. No disease symptoms typical of *Godronia* canker were observed in the plantations located in the Mazovian and Lublin Voivodships in 2019–2020. Twenty-four isolates of the *Godronia* (PIC 1 PIC 1A-PIC 1C, PIC 2, PIC 2A-PIC 2C, PIC 3, PIC 3A-PIC 3C, PIC4, PICA-PIC 4C, PIC 5, PIC 5A-PIC 5C, PIC6, PIC 6A-PIC 6C) were obtained from the infected blueberry bushes on a plantation in the West Pomeranian Voivodship only. 

### 3.2. Koch’s Postulates

After more than 3 months, the first symptoms typical for Godronia canker could be noticed on the 1.5-year-old highbush blueberry shoots inoculated with *Godronia* culture. The lesions on the shoots caused the epidermis to collapse and were dark brown to red-brown. The borders of the necrotic lesions were light brown to yellowish (Figure 1). No symptoms of infection were observed on the control plants.

### 3.3. The Fungus Morphology

Conidia on the PDA medium after 10 days had the following dimensions: 9.36 ± 0.81 µm length, 2.45 ± 0.37 µm width.

The conidia were colorless, slightly curved, ellipsoidal in shape, and mostly two-celled. They were pointed or rounded at the ends and widest, most often in the middle part (Figure 2).

### 3.4. The Pathogen Growth Dynamics

Depending on the medium type, different daily rates of colony growth of the fungal isolate were observed (Table 1, Figure 3).

The highest daily growth rate of fungal mycelium was observed on the SNA and PCA media and the lowest on CMA and MEA (Table 1).

The morphology of the cultures of *G. myrtilii* isolate on different media varied (Table 2).

The growth of the colony of the *G. myrtilli* isolate on various media is shown in Figure 4.

For all 24 fungal isolates, the amplified products obtained in the PCR reaction were identical in size. Therefore, only six isolates were selected for nucleotide sequencing. The nucleotide sequences of these isolates were identical and were 785 base pairs long. The sequence of representative isolate (PIC2) showed 100% similarity to the *G. myrtilli* reference sequence (KC595271) deposited in the GenBank. The Polish sequence of *G. myrtilli* was deposited in the NCBI GenBank under Accession number: OP902995, www.ncbi.nlm.nih.gov.genbank (accessed on 23 November 2022).

### 3.5. Phylogenetic Relationship between Godronia Isolates

Based on the prepared dendrogram, it was found that nucleotide sequences of two isolates, including the *Godronia ribis* isolates from Slovakia (MH860210) and Switzerland (MH858761), were contained within a well-supported clade (bootstrap value 100).

The remaining 11 nucleotide sequences, including the *G. myrtilli* isolate from Poland (OP902995), were contained within a second clade (bootstrap value 60). On the dendrogram, the abovementioned isolate sequence (OP902995) and the *G. cassandrae* (*G. myrtilli*) isolate sequence from Germany (KC595271) showed 100% similarity and were grouped in a separate cluster.

The *G. cassandrae* (*G. myrtilli*) nucleotide sequences from the United States of America and Canada were in a separate subgroup. The nucleotide sequence of the Polish *G. myrtilli* isolate clearly differs from that of other fungal species of the *Godronia* genus: *G. fuliginosa* or *G. ribis* (Figure 5).

## 4. Discussion

In classical phytopathological studies of various pathogens, research begins with the observation of the development of the disease (symptoms and etiological signs). The observations made during the presented studies fully confirmed the earlier findings of various Polish researchers from the last 50 years [16,18,30,39,40,41,42,43,44,45,46]. Similar results have also been obtained outside Poland and even outside Europe [4,13,24]. Differences in the timing of the appearance of the disease and/or wilting or complete shoot dieback are somewhat different in different geographical regions [3,47,48,49]. However, it is clearly emphasized everywhere that this disease is the most dangerous fungal disease of highbush blueberry [18]. In a 3-year study carried out in 3 locations in the Lublin Voivodship, Szmagara [45] has identified as many as 20 different fungi occurring in highbush blueberry plantations on the basis of the fungal structures (e.g., pycnidia) produced on shoots, with *Godronia* constituting 7.4% of all the identified fungi. 

In light of the above, it was surprising that in the 2019–2020 season, no symptoms typical of *Godronia* canker were reported in the large, productive plantations (over 50 ha) in central and southeastern Poland, where *Godronia* anamorphs (*Topospora*) have always been found. For a long time, the perfect stage (teleomorph) of *Godronia* genus has been known to be very rare and not play any role in the pathogen’s development cycle [1,20]. However, even in the mid-1970s in Poland, the apothecia of the *G. cassandrae* were sporadically found on older, symptomatic blueberry shoots (Szyndel personal comm.), with this now being almost impossible. Melzer and Hoffmann [50] have found that the pathogen has adapted to the humid, cool climatic regions of northern and central Europe. It is possible that the disappearance or nonappearance of the disease has also been influenced by very different environmental and climatic conditions of recent years, widespread use of healthy material for production plantings, new varieties, etc. This phenomenon cannot be explained only by the use of the chemical method of plant protection on production plantations.

Confirmation of the affiliation to the *Godronia* genus of all the tested isolates was done based on morphological features and nucleotide sequences. Koch’s postulates were fulfilled as well.

The conidia of *G. myrtilli* on the PDA medium had very similar dimensions (9.36 ± 0.81 × 2.45 ± 0.37 µm) to those depicted by Groves [20]. On the other hand, in the studies conducted on the same medium by Melzer and Hoffmann [50], the average length of the conidia was greater and ranged from 10.82 to 12.47 µm. In the studies of Weber and Entrop [51], the conidia were 10.7–16.7 µm long and 1.7–2.7 µm wide, while in the study by Weingertner [52], the dimensions of the conidia were different depending on the cultivar, e.g., on Earliblue, the dimensions were 11.2 × 1.4 µm, and on Jersey, 12.2 × 1.6 µm. The morphology of the conidia described in the abovementioned reports is very similar to that described in the presented paper.

In the literature, there is a lack of information on the growth dynamics of the pathogen on different media; only data on the linear growth of the *Godronia* cultures are available. The fastest linear growth of the fungus was noted on the PDA medium and slightly slower on MEA [25], or the fastest linear growth was on the MEA medium and slower on the CMA medium [16]. The morphology of the isolates on different media varied. PDA appears to be the best medium for the growth of *G. myrtilli* colonies, even though it does not have the fastest fungal growth rate. However, fungal colonies on this medium have a very well-developed aerial mycelium, and pycnidia formation occurs. The presented research indicated that organic media are more suitable for testing this pathogen than nutrient-poor media. It was also confirmed by Borecki and Pliszka [16] and Szmagara and Zalewska [18]. The PDA medium was most often used in studies on the morphology and etiology of *G. cassandrae* [2,3,20,50].

Molecular research on fungi of the *Godronia* genus is still not very advanced, with few scientific reports available [50,53]. Currently, nucleotide sequences of only 42 *G. cassandrae* (*G. myrtilli*) isolates are deposited in the GenBank [54]. In Poland, molecular characterization of fungus isolates was performed for the first time in this study. According to the created dendrogram, all nucleotide sequences of the tested isolates are very similar, although the European isolates belong to a different subgroup than the American isolates. On the other hand, the nucleotide sequences of *G. cassandrae* (*G. myrtilli)* are significantly different from those of other fungal species of the *Godronia* genus.

## 5. Conclusions

Based on the obtained results, it can be stated that shoot dieback of highbush blueberry was rarely caused by *G. myrtilli* due to a sporadic occurrence of the pathogen.The growth of *G. myrtilli* depends on the medium type. The most intense fungal growth was observed on the SNA and PCA media, and the lowest on CMA and MEA.The nucleotide sequences of the fungal isolates tested were very similar to those originating from Europe.

## Figures and Tables

**Figure 1 pathogens-12-00642-f001:**
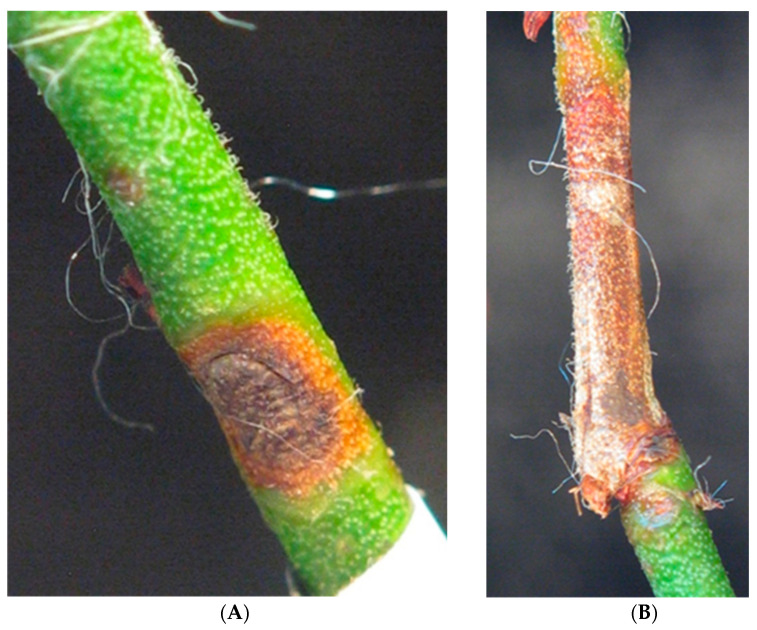
(**A**) Necrotic lesions on the stem of highbush blueberry cv. “Bluegold” artificially inoculated with *G. myrtilli* (isolate PIC3D) in Koch’s postulates test (**B**) replication 1 (photo Mirzwa-Mróz E.).

**Figure 2 pathogens-12-00642-f002:**
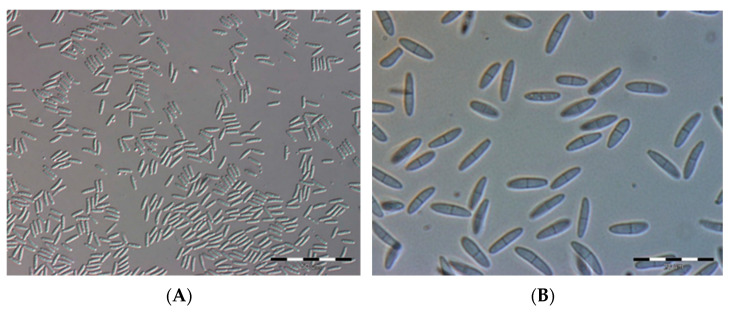
(**A**) Conidia of *G. myrtilli* on the PDA medium (photo Kuźma K.), (**B**) magnification (40×) (photo Wdowiak M.).

**Figure 3 pathogens-12-00642-f003:**
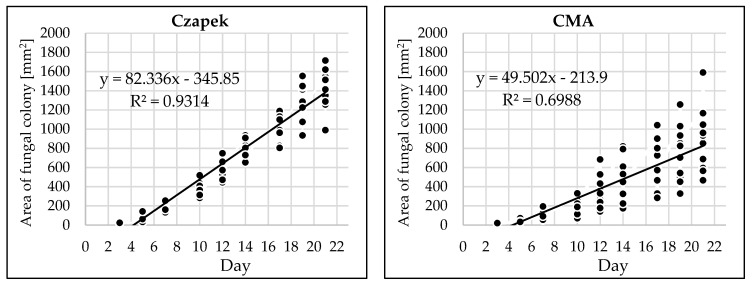
Growth dynamics of *G. myrtilli* on six media.

**Figure 4 pathogens-12-00642-f004:**
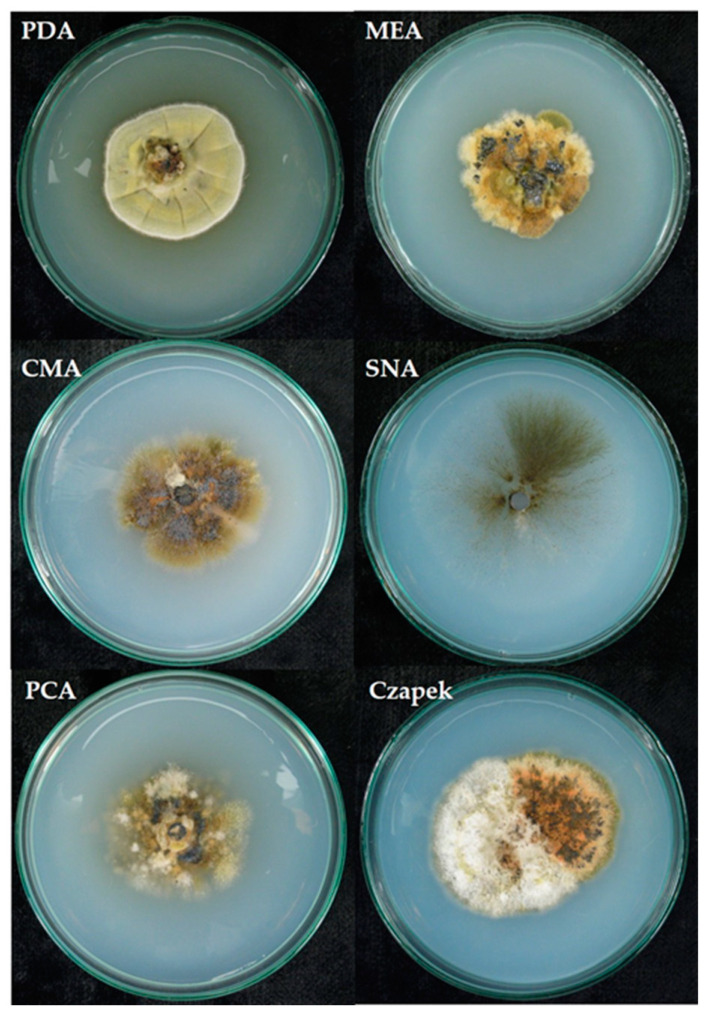
Growth of the *G. myrtilli* colony (isolate PIC2) on various media (photo Mirzwa-Mróz E.).

**Figure 5 pathogens-12-00642-f005:**
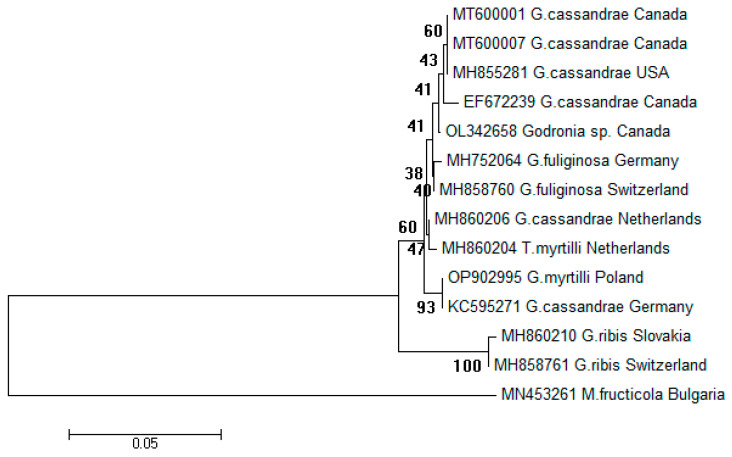
Phylogenetic tree for *Godronia* spp. isolates based on a small subunit ribosomal RNA gene, partial sequences; an internal transcribed spacer 1 and 5.8S ribosomal RNA gene, complete sequence; and an internal transcribed spacer 2, partial sequences. As an outgroup, *M. fructicola* strain was included. Branch length values are shown; the tree was reconstructed using the maximum likelihood approach and tested by bootstrapping (1.000 replicates).

**Table 1 pathogens-12-00642-t001:** The daily growth rate (mm^2^/day) of *G. myrtilli* (b coefficient) on different media.

Isolate	Medium	b Coefficient of the Regression Equation	Homogeneous Groups *
PIC2	MEA	42.5	a
	CMA	49.5	a
	PDA	75.3	b
	Czapek	82.3	b
	PCA	105.4	c
	SNA	121.8	c

* Homogeneous groups according to the Student-Newman-Keuls test, *p* = 0.05.

**Table 2 pathogens-12-00642-t002:** Morphology of *G. myrtilii* cultures on different media.

Medium	Descriptions of Fungal Colony Morphology
PDA	Colony well-developed, fluffy with radial depressions, slightly folded, slightly raised, yellowish-greenish to gray-olive; the margin of the colony is undulating, white-gray, and the reverse is brown-black.
MEA	Colony erect, folded, well-developed, black through olive to brown; olive-brown reverse, wavy margin; pycnidia (3 weeks) with one-celled conidia without formed septa.
CMA	The colony is folded, slightly raised, with a lobed margin, similar to the reverse, brick-brown; poorly developed aerial mycelium; single black structures visible in the medium.
SNA	Mycelium poorly developed, gray, with clearly visible hyphae growing radially into the substrate; the margin of the colony is barely visible, undulating; gray-brown reverse; mucoid droplets containing two-celled conidial spores formed on the surface of the medium.
PCA	Colony erect, folded, well-developed, mycelium divided into segments, white-gray-olive, black-rusty-olive, olive reverse, black structures formed in the medium.
Czapek	Fluffy mycelium, abundantly developed, raised; white-gray-olive colony, black-rusty-olive segments clearly visible in some sites of the colony; the margin of the colony is undulating, and the reverse is olive-gray-brown.

## Data Availability

Not applicable.

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
