# Peer review of "Phenotypic Characterization and Phylogeny of Godronia myrtilli (Anamorph: Topospora myrtilli)—Causal Agent of Godronia Canker on Highbush Blueberry"

_pathogens, 2023, doi:10.3390/pathogens12050642_

Round 1
Reviewer 1 Report
Dear Authors,
I found some lack of italics in fungal Latin names (mostly, when the genus names were used), they are marked in yellow. Also, one suggestion for the sentence I have added to improve understanding of the whole sentence.
My suggestions to improve the text are minute and do not influence the whole text, which presents a good quality.

Author Response
Dear Reviewer
On behalf of the Authors I would like to thank you for your valuable comments and suggestions. We have taken them into account and revised our manuscript accordingly.
The changes in the manuscript we marked up using the “Track Changes” function.
Best regards
Ewa Mirzwa-Mróz
Reviewer 2 Report
The original manuscript consists of 12 pages including Tables, Figures and References and present new information and discussing about phenotypic characterization and phylogenetic analysis of Godronia spp. fungi.
The authors should present the materials and methods in subsections depending on the experimental processes or according to the presentation of the result.
Line 65. Maybe it would be appropriate to indicate the coordinates or explain geographical/ meteorological plant growing conditions; or is this appropriate for highbush blueberry growth anywhere.
Lines 89-91. The authors written about 6 media, but really are seven “CMA (corn meal agar), PDA (potato dextrose agar), MEA (malt extract agar), WA (water agar), SNA (synthetic nutrient-poor agar), PCA (potato carrot agar) and Czapek (Czapek solution agar)”. What happened to water agar?
Lines 99-103. The formula is not fully explained.
Lines 242-247.This part of the discussion is not completely correct because the authors do not provide any information about meteorological conditions or chemical control of plant in the article.
Author Response
Dear Reviewer
Thank you very much for your valuable comments and suggestions. We have taken them into account and revised our manuscript accordingly. The changes in the manuscript we marked up using the “Track Changes” function. The replies to the comments you can find below:
1)The authors should present the materials and methods in subsections depending on the experimental processes or according to the presentation of the result.
We have divided Materials and Methods according to your suggestions.
2)Line 65. Maybe it would be appropriate to indicate the coordinates or explain geographical/ meteorological plant growing conditions; or is this appropriate for highbush blueberry growth anywhere.
Meteorological data were not included in the manuscript, as we did not carried out studies on the epidemiology of the disease. The investigations presented in current paper were preliminary and are the base for the deeper study in the future. The inspection of the plantations were conducted in order to obtain isolates of the fungus and further characterize them. We understand that epidemiology of Godronia canker is a very interesting issue and research on it will be conducted in the future.
3)Lines 89-91. The authors written about 6 media, but really are seven “CMA (corn meal agar), PDA (potato dextrose agar), MEA (malt extract agar), WA (water agar), SNA (synthetic nutrient-poor agar), PCA (potato carrot agar) and Czapek (Czapek solution agar)”. What happened to water agar?
Experiments were carried out on 6 media according to Table 1, Fig. 3 and Tab. 2. By mistake in Material and Method I included WA (Water Agar). I have delete it from the text.
4)Lines 99-103. The formula is not fully explained.
We have added formula description according to your suggestions.
5)Lines 242-247.This part of the discussion is not completely correct because the authors do not provide any information about meteorological conditions or chemical control of plant in the article.
In the discussion, we did not refer to our research results on this issue, and only based on the literature we expressed our assumptions, which in our opinion could be the reason for the lack of this disease on highbush blueberry plantations.
On behalf of the Authors
Ewa Mirzwa-Mróz